# Microwave Imaging Approach for Breast Cancer Detection Using a Tapered Slot Antenna Loaded with Parasitic Components

**DOI:** 10.3390/ma16041496

**Published:** 2023-02-10

**Authors:** Fatima-ezzahra Zerrad, Mohamed Taouzari, El Mostafa Makroum, Jamal El Aoufi, Salah D. Qanadli, Muharrem Karaaslan, Ahmed Jamal Abdullah Al-Gburi, Zahriladha Zakaria

**Affiliations:** 1Laboratory IMII, Faculty of Sciences and Techniques, Hassan First University of Settat, Settat 26000, Morocco; 2Laboratory LISA, National School of Applied Sciences, Hassan First University of Settat, Berrechid 26100, Morocco; 3Laboratory of Aeronautical & Telecommunication, Mohammed VI, International Academy of Civil Aviation, Casablanca 20000, Morocco; 4Department of Diagnostic and Interventional Radiology, Lausanne University Hospital, 1011 Lausanne, Switzerland; 5Electrical-Electronics Engineering, Iskenderun Technical University, 31200 İskenderun, Turkey; 6Center for Telecommunication Research & Innovation (CeTRI), Fakulti Kejuruteraan Elektronik dan Kejuruteraan Komputer (FKEKK), Universiti Teknikal Malaysia Melaka (UTeM), Durian Tungal, Malacca 76100, Malaysia

**Keywords:** microwave imaging, UWB, parasitic element, microstrip antenna, breast tumor

## Abstract

In this paper, a wideband antenna is proposed for ultra-wideband microwave imaging applications. The antenna is comprised of a tapered slot ground, a rectangular slotted patch and four star-shaped parasitic components. The added slotted patch is shown to be effective in improving the bandwidth and gain. The proposed antenna system provides a realized gain of 6 dBi, an efficiency of around 80% on the radiation bandwidth, and a wide impedance bandwidth (S11 < −10 dB) of 6.3 GHz (from 3.8 to 10.1 GHz). This supports a true wideband operation. Furthermore, the fidelity factor for face-to-face (FtF) direction is 91.6%, and for side by side (SbS) is 91.2%. This proves the excellent directionality and less signal distortion of the designed antenna. These high figures establish the potential use of the proposed antenna for imaging. A heterogeneous breast phantom with dielectric characteristics identical to actual breast tissue with the presence of tumors was constructed for experimental validation. An antenna array of the proposed antenna element was situated over an artificial breast to collect reflected and transmitted waves for tumor characterization. Finally, an imaging algorithm was used to process the retrieved data to recreate the image in order to detect the undesirable tumor object inside the breast phantom.

## 1. Introduction

Breast cancer is now a leading cause of death in women all over the world [1]. It is a difficult disease to prevent because it grows in such a complex manner, involving several cells. As depicted in Figure 1, 9.2 million cancer cases were recorded among women in 2020 [2]. However, an accurate diagnosis of breast cancer can considerably lower the likelihood of it progressing [3]. Due to early detection and treatment, breast cancer patients in certain developed countries have a 5-year survival rate of more than 80% [4]. Various imaging techniques, such as ultrasound, X-ray, computed tomography (CT), magnetic resonance imaging (MRI), and nuclear medicine imaging, have been developed in recent decades for the identification of breast tumors. Microwave imaging (MWI) is currently a promising approach for detecting anomalies in the human body based on the electrical properties of both healthy and malignant tissue. Moreover, MWI is distinguished from other medical imaging methods due to its simplicity, low cost, and lack of ionizing radiation [5]. The relative permittivity and conductivity of biological tissue are the most important dielectric properties, with the latter being inextricably linked to the amount of water present in the tissue sample. Due to the dielectric characteristics of malignant and healthy breast tissue at microwave frequencies, MWI devices have been researched in recent years for early-stage breast cancer diagnostics [6,7,8]. Cancerous tissue differs from healthy tissue because the permeability of the cancer cell membrane changes, allowing more water to pass into the cell [9]. As a result, malignant cells have more water and dissolved ions inside them with respect to healthy cells of exactly the same type of tissue. This results in higher relative permittivity and conductivity values that can interact differently with the incident microwave compared to the healthy cells, leading to successful detection.

One of the most critical elements of the MWI system is the antenna, which is responsible for transmitting and receiving the electromagnetic energy required to generate the image. The antennae utilized in this type of biomedical imaging system should cover as much of the microwave medical imaging systems’ frequency band as possible while still being compact to fit in reasonably sized imaging systems. Ultra-wideband (UWB) antennae satisfy this requirement since they are compact and cover a considerable portion of the frequency range required in microwave medical imaging systems [10]. They should operate between 3.1 GHz and 10.6 GHz according to the specifications in [11].

In recent years, a number of UWB antennae for microwave imaging systems (MIS), in which good qualities of both wave penetration into tissue and spatial resolution can be achieved, have been developed: in [12] slotted antennae [11], coplanar waveguide (CPW) antennae [13], Vivaldi antennae [14,15], metal-backed artificial magnetic conductors (AMC) [16], textile-based antennae [17], metamaterial (MTM) antennae [18] and resistively loaded bowtie antennae.

This research aims to contribute to the technological advancement of breast cancer detection systems by focusing for the first time in the literature on the construction of a UWB-slotted antenna with four star-shaped parasitic elements for detection systems.

Moreover, this study presents a breast tomography system for observing breast health and early monitoring of malignancies. The system employs two novel UWB antennae that operate as a transmitter and a receiver.

The paper is organized as follows. Section 2 includes the antenna design considerations. The impacts of various geometrical parameters on antenna performance are also described in Section 2, along with numerical data. Furthermore, prior to adopting the proposed antenna-driven breast imaging system inquiry, a realistic heterogeneous lab-made breast phantom is constructed and successfully tested in Section 3. The performance of imaging of the module for artificial breast is discussed in Section 4. The final section, Section 5, provides the conclusion.

## 2. Antenna Design

Figure 2 depicts the proposed antenna design. The overall dimensions of the antenna are shown in the same figure. The overall antenna size is 29 × 26.6 × 1.575 mm^3^. The feed and one parasitic star resonator are printed on the bottom side of a Rogers RT/Duroid 5880 substrate material, with a dielectric constant of 2.2 and a loss tangent of 0.0009, as depicted in Figure 2A. The feed dimensions are optimized to match a conventional 50 ohm feed line (coaxial signal cable). The feed shape was selected to support broadband operation. The star patch near the feed microstrip line was found useful in achieving better matching to the feed line. On the top side of the substrate, a tapered slot antenna comprised of a rectangular slot topped by a triangle was engraved within the metallic ground plan, as shown in Figure 2B. Three star-shaped metallic resonators are left inside the slot for better matching and for improved gain.

The dimensions, sizes and positions of the star resonators have been depicted in Figure 2. The appropriate sizes are determined by utilizing a parametric CST microwave study. In this process, the dimensions of the stars have been assigned as variable. The selection of the suitable dimensions has been realized for the previously determined resonance frequency. The parametric study has been carried out using a genetic algorithm approach to response around resonance frequency. The limitations of the stars’ dimensions are defined as not to contact with the main antenna structure and to stay at the level of the port input. The physical response of the stars has been demonstrated as current distribution at resonance frequencies. In addition, the position of the star has been determined to interact simultaneously with both the main line and one side line. It is expected to improve capacitive effect and inductive–capacitive matching to provide resonance.

Impedance matching is shown to be sensitive to the presence of parasitic resonators, a rectangular slotted radiating patch element, and a tapered slot ground. Moreover, the level of the undesired lobe is reduced as the main lobe gain is increased, correcting squint effects and increasing radiation characteristics by adding the star-shaped parasitic components [19]. Figure 3A depicts the effects of the parasitic resonator on the reflection value. It shows that the resonators denoted as parasitic elements have a considerable effect on improving bandwidth significantly between 7 and 8 GHz. Figure 3A also reveals that these star resonators also enhance the s11 bandwidth by expanding the <−10 S11 a further 600 MHz (shifted from 4.22 to 3.88 GHz). On the other hand, Figure 3B depicts how the gain of the antenna changes based on the design. The proposed antenna gain is higher when the parasitic star resonators are used. In fact, the maximum gain is enhanced by 0.3 dBi to reach 6.8 dBi by employing parasitic resonators. Additionally, when parasitic components are employed, different current conducting paths are created. Hence, the input impedance of the antenna system composed of the capacitance and inductance is changed, causing the antenna characteristics to change.

Figure 4A,B shows the surface current distribution of the proposed antenna at 4.5 and 7.5 GHz, respectively. The feed strip, mainly at the electric current application port, is the most essential surface current conducting zone of the proposed antenna. Since the flow of electrons is nonlinear at higher frequencies and the primary current conducts around the patch, the stability of the distribution of current is better at a lower frequency with respect to the higher one. Furthermore, there is a significant current flow near the cutting plane’s edge at a low frequency. The parasitic element, on the other hand, affects the metallic path and antenna characteristics.

## 3. Results and Discussion

Before going through the microwave imaging results, it is important to understand how the proposed UWB antenna works in both the temporal and frequency domains.

Generally, narrow-band antennae and propagation are usually defined in terms of frequency. The characteristic parameters are usually believed to be constant over a few percentage points of bandwidth. The frequency-dependent features of the antennae and the frequency-dependent behavior of the channel should be studied for ultra-wideband (UWB) systems. However, UWB systems are frequently achieved by using an impulse-based technology; thus, the time-domain impacts and characteristics should be understood as well. Otherwise, the transmitter antenna is stimulated with a continuous wave for the frequency-domain description. The transmitter antenna is stimulated with an impulse signal of frequency f for the time-domain description.

### 3.1. Frequency Domain

To test the results of numerical analysis, a prototype of the simulated antenna is built utilizing a CNC-based PCB device (Figure 5A). The S11 value is determined by using VNA. Figure 5B–D shows the fabricated prototype and measuring system.

Figure 5E shows the antenna’s simulation and measurement reflection value (S11) results according to the frequency. There is a small difference between the two values due to manufacturer tolerance and improper soldering of the SMA connector. For S11 below -10 dB, the simulated impedance BW is 6.3 GHz. The measurement and simulation values are in good agreement throughout the BW.

Figure 6A displays the Geozondas antenna measurement system (GAMS) that is used. A horn antenna is chosen as transmitter to measure the realized gain and the radiation pattern of the antenna, while the suggested antenna is operated as a receiver.

Figure 6B demonstrates the gain and efficiency variation as a function of frequency.

As a result, the simulated antenna has a maximum gain of 6.8 dBi at 10 GHz and great efficiency of 96% at 8 GHz, indicating that the majority of the power at the antenna’s input is radiated out effectively. In addition, the proposed antenna has a high realized gain and efficiency when compared to the other antennae.

Figure 7A,B displays the E and H planes in two different planes, as well as co- and cross-polarization for two frequencies of 4.5 GHz and 7.5 GHz. The antenna transmits electromagnetic waves omnidirectionally in the E and H planes, as shown in Figure 7. At a low frequency of 4.55 GHz, radiation patterns with co-polarization of the E plane are slightly directed.

The radiation performance of the antenna system has been selected as almost isotropic. This is related to the selection of the number of antennae in the system. Since the planned antennae to be utilized in the system must be three receivers and one transmitter, the transmitting antenna must radiate to each receiver with almost equal magnitude. Hence, all antennae must transmit and receive signals in a wide angle. The selection of four antennae stems from the measurement facility of the laboratory. The increment in antenna numbers and reducing of the beamwidth will be realized in future studies according to the measurement facility potentials.

The PSG signal generator (SG) and the CXA signal analyzer (SA) n9000b 9 kHz–26.5 GHz from Keysight are used to extract the proposed antenna characteristics. The measurements involving the SG, the planned antenna, and the SA are carried out using the following approach, where the air is the propagation medium. As shown in Figure 8A,B, the output power of the SG transmitter is measured across the FtF and SbS scenarios of the antenna, respectively.

The SG is used to transmit a pulsed CW RF signal to a coaxial cable-coupled TX antenna. As a result, the RX antenna receives the TX signal, and SA measures the transmitted power.

The frequency response of a 13 dBm pulsed CW RF signal irradiated by the SG at 7.5 GHz is shown in Figure 9. The frequency-domain response shows a noise floor of roughly −77 dBm without the SG pulse signal. Since the signals are affected by the environment in the case of tests carried out in free space, after the pulse, the received signals of −19.8 dBm and −23.8 dBm are measured for FtR and SbS, respectively.

### 3.2. Time-Domain Performance

The antenna transfer functions describe a two-dimensional vector with two orthogonal polarization components in the frequency domain. However, the antenna’s transient response becomes more appropriate for the description of impulse systems in the time domain. The antenna’s transient response is affected by time, as well as departure, arrival angles and polarization.

Figure 10 depicts the time-domain method for sending and receiving a pulse between two antennae.

Figure 11A,B demonstrates the incident and transmitted waves propagating along two orientations with a length of 250 mm between the two suggested radiators. TX and RX signal values show that orientation of FtF and SbS have low reduction of transmitted signals.

In addition, the highest value of cross-correlation between the transferred and gathered signals presents the fidelity factor. The fidelity factor is evaluated [17].

The fidelity factors are 91.6% and 91.2% for the orientations of FtF and SbS, respectively. According to these results, the fidelity factor of the suggested antenna in the case of FtF is higher than SbS. The better fidelity factor provides lower reduction of the transmitted wave. This property indicates that the antenna is appropriate in microwave imaging applications.

The group delay (τ) is the retardation of time where the wave transfers from the transmitter to the receiver point. Figure 12 presents the effects of the parasitic resonator on the group delay in face-to-face and side-by-side scenarios.

According to Figure 12, the parasitic resonators have a significant improvement in reducing the group delay in both scenarios.

Figure 13 presents the change of (τ) in terms of frequency for FtF and SbS positions.

It must be mentioned that the group delay is numerically analyzed for both types of orientation in the case of 250 mm distance between the antennae. The change in group delay throughout the entire operating band is below ±2.0 ns [20], which can be defined as an acceptable value for MWI studies. The figure of the group delay for face orientation is nearly linearly distributed, which acts to transmit short waves to the antenna with lower late-time ringing and distortion. Hence, it is proposed to utilize the suggested antenna in an FtF orientation.

## 4. Microwave Imaging

### 4.1. Fabrication of Heterogeneous Breast Phantom

The usefulness of MWI breast cancer detection systems can be evaluated using tissue-imitating materials prior to clinical experiments, with numerous ways employed to construct a realistic breast phantom [18].

Despite this, there is still no consensus or standardization in the field as to which form of lab-made breast phantom is appropriate for MWI system investigations. As a result, one of the goals of this research is to construct and test the proposed antenna’s performance using a realistic breast phantom.

This part focuses on the creation of a breast phantom, which includes simulation and production techniques, as well as the usage of materials to simulate the diverse breast tissue composition found in vivo. The simulated and constructed breast phantom in this work is created using the dielectric characteristics indicated in Table 1.

The design of the three-layered heterogeneous breast phantom model is shown in Figure 14. Figure 14A shows the simulated breast phantom model for analysis. Furthermore, in both simulation and fabrication, a tumor with a diameter of 10 mm is examined. Figure 14B displays a breast model with R1 = 50 mm, R2 = 48 mm, and R3 = 30 mm with a height of 30 mm.

As shown in Table 2, there is a similar preparation phase for each tissue type in the breast phantom, with varied material concentrations. This section shows a 3D-printed breast phantom created from the anatomy of a genuine human breast using the CubePro Trio 3D printer.

The steps needed in making the breast phantom are depicted in Figure 15.

Step 1: Measure the quantity of material according to Table 2.Step 2: Add distilled water, oil and P.P.J to flour.Step 3: Add NaCl.Step 4: Add powder.

Figure 16A depicts the assembled components. The dielectric properties of a tissue-mimicking breast phantom are determined by using an open-end coaxial connected probe kit and Agilent PNA-L Vector Network Analyzer (VNA). The Agilent 85070E dielectric probe kit contains a coaxial probe (shown in Figure 16B) that can measure relative dielectric constants from 200 MHz to 20 GHz frequency.

Figure 17 shows the measured dielectric constant and conductivity of each layer as a function of frequency. Following the preparation of each layer, the final fabrication technique of the proposed breast phantom is shown in Figure 18 in a step-by-step arrangement.

### 4.2. Microwave Imaging Simulated Setup

The proposed antenna will be employed as a radiating element in an MWI-based breast cancer detection system. Figure 19 depicts a configuration of four antennae surrounding a breast model for testing antenna performance in an imaging system.

Figure 20 depicts simulated imaging results of the electric field of the prototype, shown in Figure 19, for breasts with and without tumors to better comprehend the changes in signal scattering. Antenna 1 is the only one that emits radiation in this scenario. In fact, microwaves behave differently when they come into interface with tissue with differing dielectric properties. The signal scattering percentage inside normal tissue is larger than that of tumor cells, because in comparison to normal breast tissue, tumor cells have higher dielectric properties. As a result, the tumor cell reflects more signal. The scatter signals from the breast tissue and the malignant tissue also show a significant difference in the imaging results. Additionally, a high-resolution image that shows the location of the tumor cell has been highlighted in red.

### 4.3. Microwave Imaging Experimental Setup

As shown in Figure 21, an experimental MWI is conducted with a constructed breast phantom and the proposed antennae, and data are obtained with this configuration. The scattering signals are generated by the interference between microwave signals from antennae and the tissue of the breast, which can be altered by path propagation speed, phase, polarization, and surface strength [21]. For any particular measurement, one antenna emits, and multiple antennae receive in a multi-static setup, and entire reflected and transmitted waves are collected. In this work, in addition, the phantom is rotated at 30° to achieve the best results with this multi-static antenna system, as shown in Figure 21A. A board is marked appropriately throughout 0–360 degrees (Figure 21B) with a 30-degree range and then placed on a rotating table. Each antenna is kept at a distance of 15 mm from the breast structure (Figure 21C). The reflected data are then obtained for future investigation.

The microwave pulse penetrates the phantom, with electromagnetic waves bouncing back from different layers of tissue. MATLAB-based MERIT (Microwave Radar Based Imaging Toolbox) [22] software (R2021a) is used to record and further analyze backscattered data.

Figure 22 shows the revised image depending on the experimental study, with the high-contrast zone representing the tumor’s center. The tumor object has been clearly identified with red, as can be seen. However, the malignancy appeared somewhat off-center on the image compared to its actual location. This suggests that our technique could be a promising candidate for microwave imaging in order to detect undesired cell-like tumors by efficiently evaluating backscattering signals.

Table 3 shows a comparison of reported and proposed antennae. BW, dimension, antenna gain, and setup realization are the parameters that are examined. The suggested antenna features a smaller compact, a higher gain, and a larger bandwidth than the previously reported antennae.

## 5. Conclusions

This article summarizes the imaging performance of ultra-wideband antennae to detect breast tumors. The suggested antenna, according to the results, has a wide BW of 6.3 GHz (3.8–10.1 GHz). Furthermore, a high transmission effectiveness over 80% and a gain of 6.8 dBi are achieved. The antenna is a promising candidate for microwave imaging studies due to its basic structure, compact characteristics, and ultra-wideband properties. Furthermore, the antenna operates with high efficiency in both the frequency and time domains, with fidelity factors of 91.6% in FtF and 91.2% in SbS. A configuration including an artificial breast is investigated at the end of the study, and the antennae are properly placed and identify the existence of the tumor inside the phantom breast.

## Figures and Tables

**Figure 1 materials-16-01496-f001:**
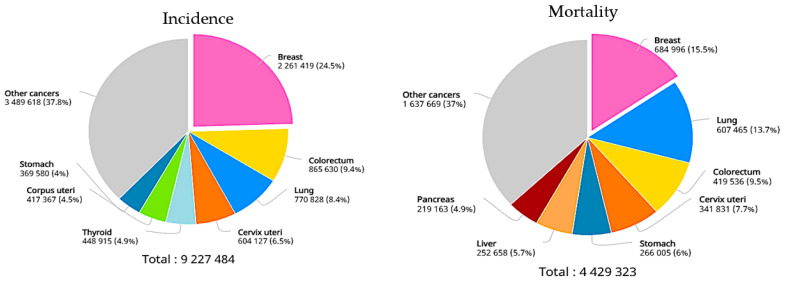
Approximate number of yearly cancer cases/deaths in 2020 [2].

**Figure 2 materials-16-01496-f002:**
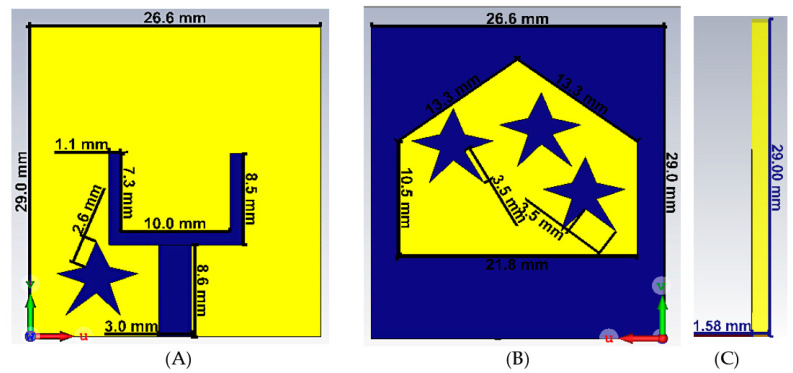
The proposed antenna model. (**A**) Bottom view, (**B**) top view, and (**C**) side view.

**Figure 3 materials-16-01496-f003:**
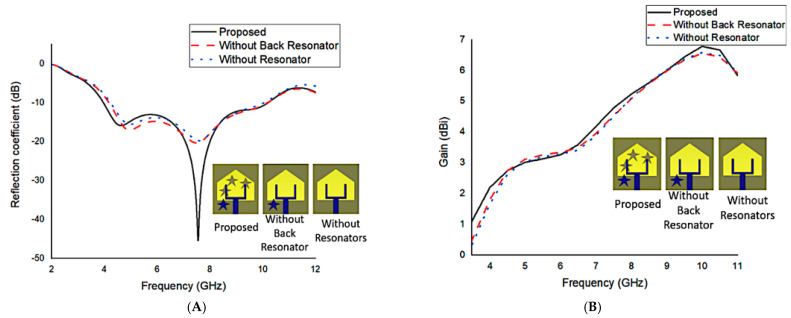
The parasitic resonator’s effects on the reflection coefficient (**A**) and the gain (**B**).

**Figure 4 materials-16-01496-f004:**
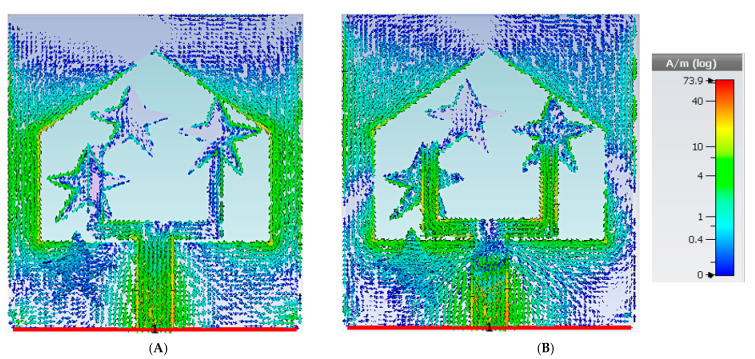
Current allocation at (**A**) 4.5 GHz and (**B**) 7.5 GHz.

**Figure 5 materials-16-01496-f005:**
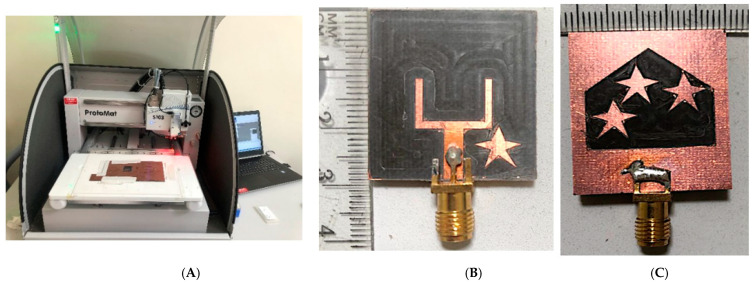
(**A**–**D**) The fabricated prototype and measuring system and (**E**) the antenna’s simulated and measured S11.

**Figure 6 materials-16-01496-f006:**
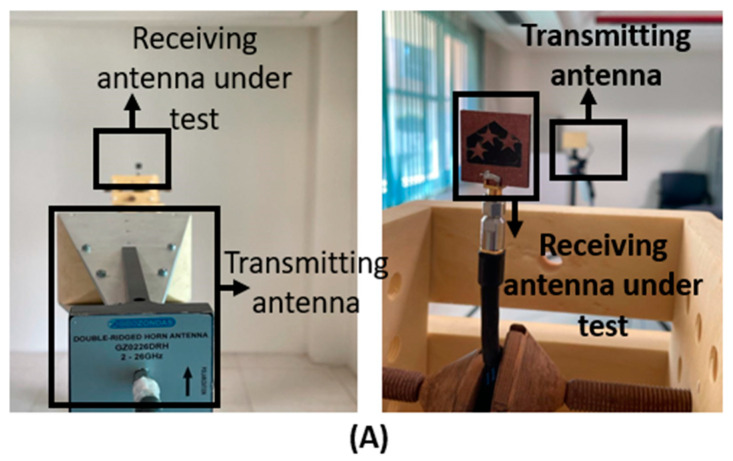
(**A**) Geozondas measurement system, (**B**) realized gain, and (**C**) efficiency.

**Figure 7 materials-16-01496-f007:**
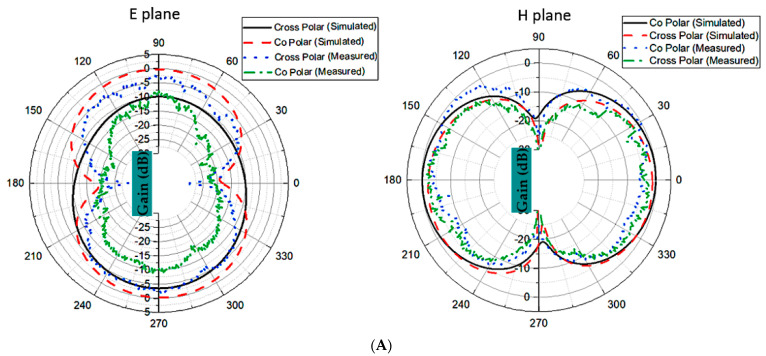
Results of E and H plane at (**A**) 4.5 GHz, and (**B**) 7.5 GHz.

**Figure 8 materials-16-01496-f008:**
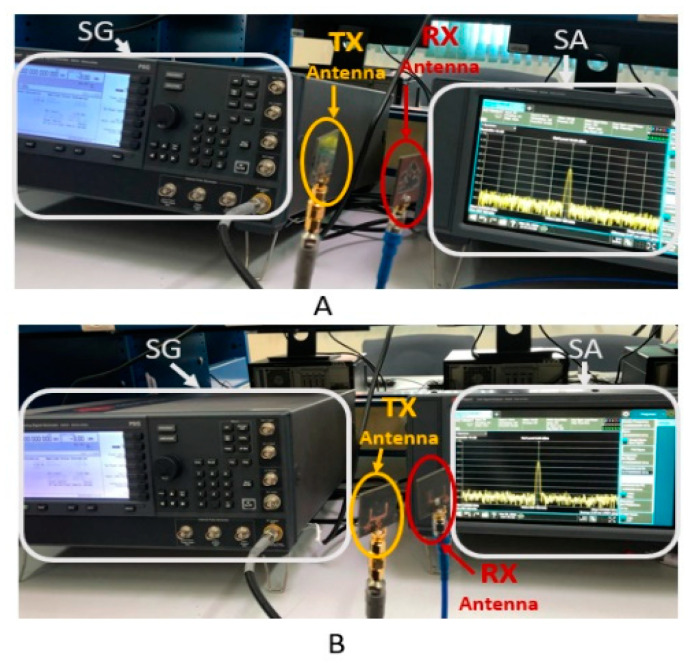
Signal generator transmitting to spectrum analyzer via TX and RX antennae (**A**) for FtF and (**B**) for SbS scenarios.

**Figure 9 materials-16-01496-f009:**
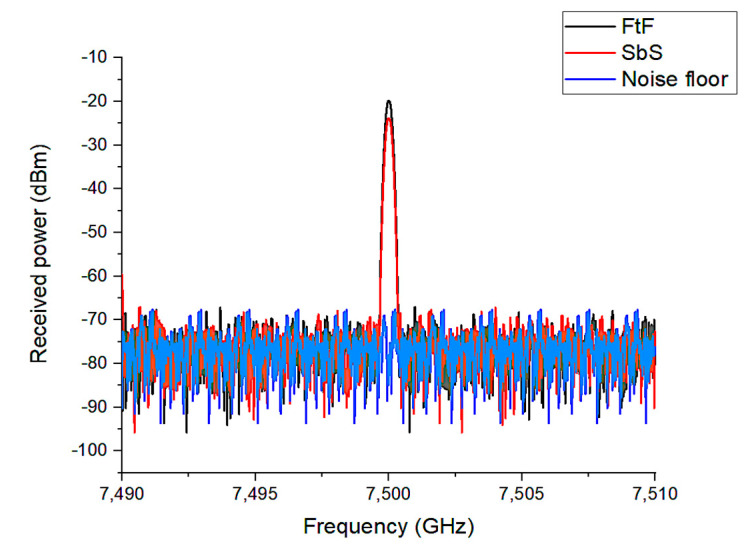
Measured RF power for FtF and for SbS scenarios.

**Figure 10 materials-16-01496-f010:**
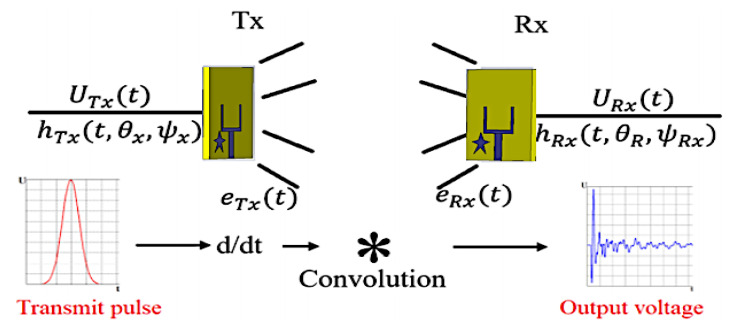
Time-domain method for sending and receiving a pulse between two antennae.

**Figure 11 materials-16-01496-f011:**
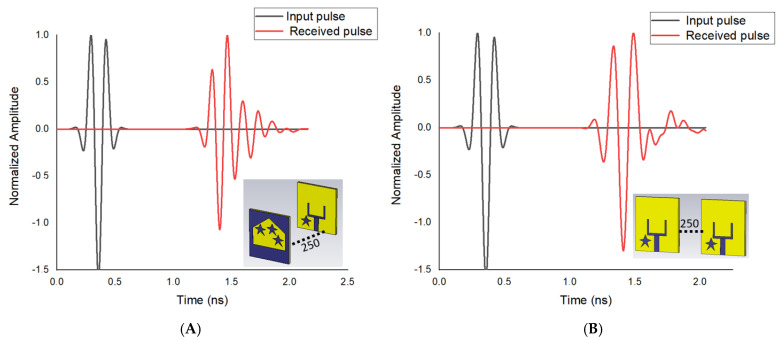
Input and received pulse in (**A**) FtF, and (**B**) SbS.

**Figure 12 materials-16-01496-f012:**
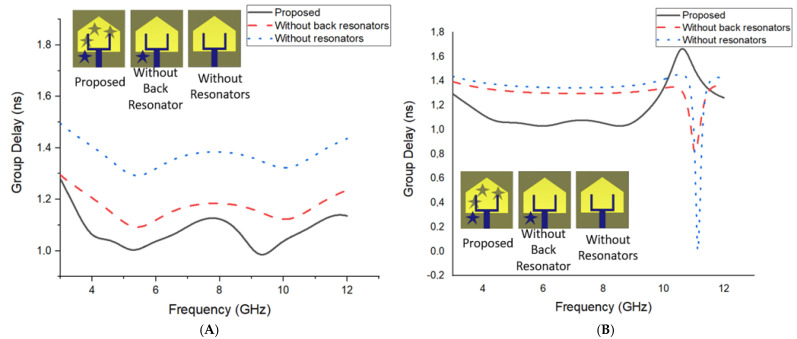
The parasitic resonator’s effects on group delay (**A**) face to face and (**B**) side by side.

**Figure 13 materials-16-01496-f013:**
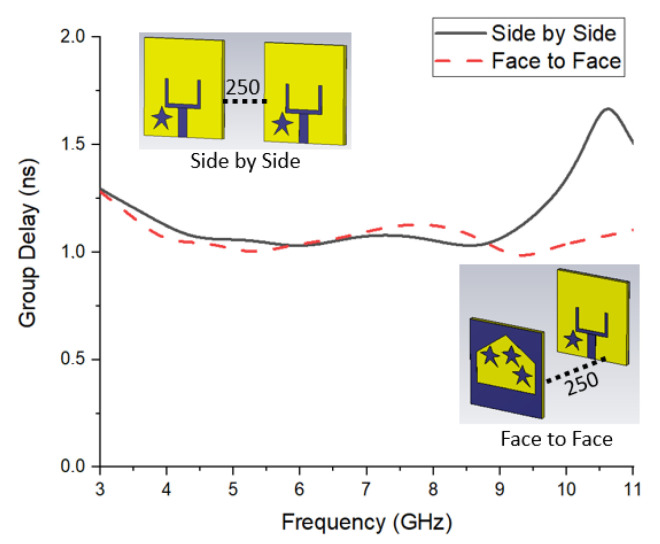
Group delay variation for different orientations of antennae.

**Figure 14 materials-16-01496-f014:**
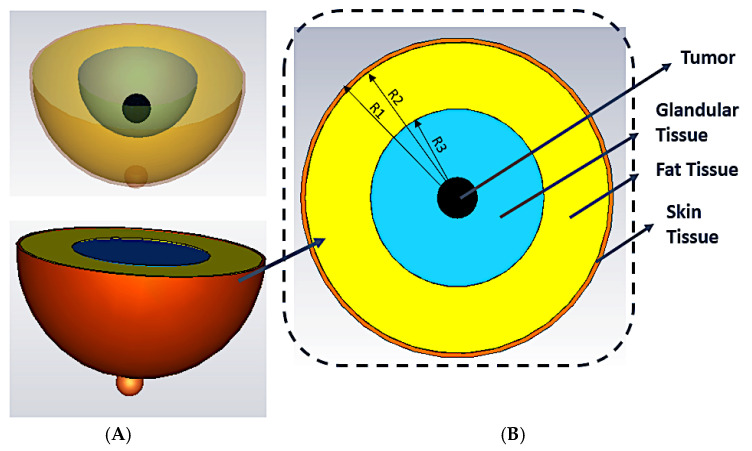
(**A**) The breast model from the top and (**B**) breast phantom model in 3D simulation.

**Figure 15 materials-16-01496-f015:**
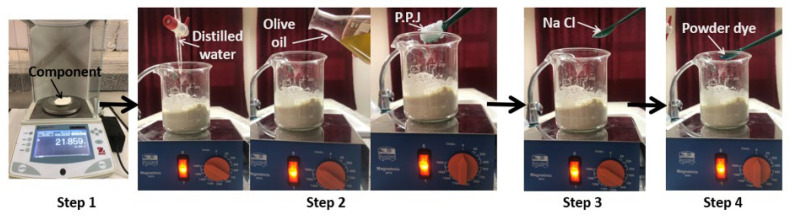
Procedure for making a heterogeneous breast phantom [18].

**Figure 16 materials-16-01496-f016:**
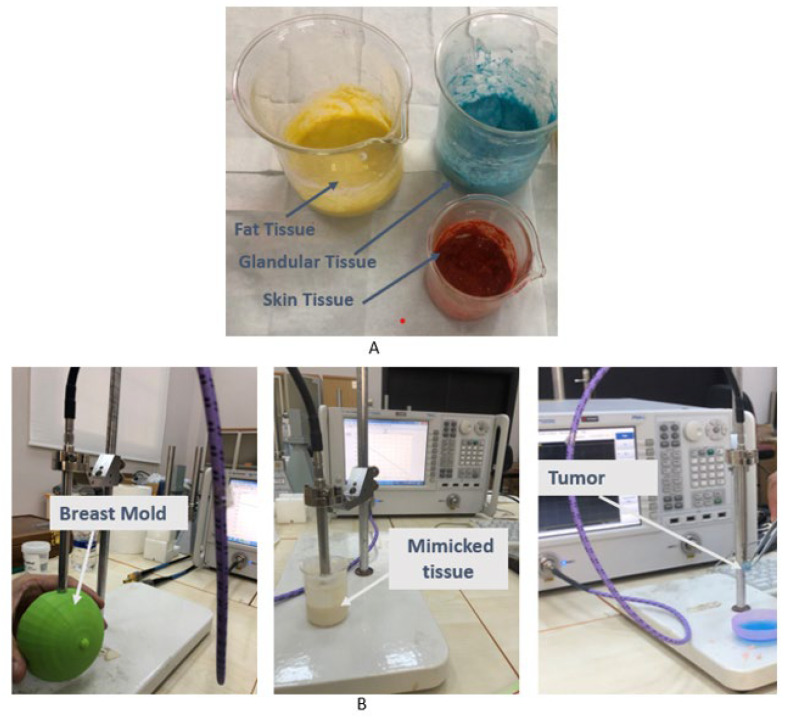
(**A**) The fabricated breast tissue components and (**B**) breast phantom dielectric measurement.

**Figure 17 materials-16-01496-f017:**
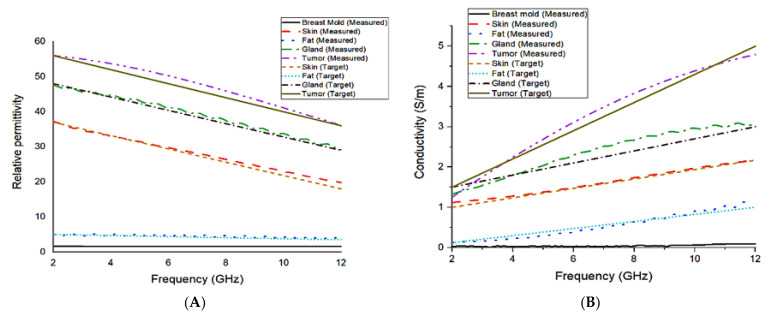
(**A**) Measured and compared relative permittivity and (**B**) measured and compared conductivity.

**Figure 18 materials-16-01496-f018:**
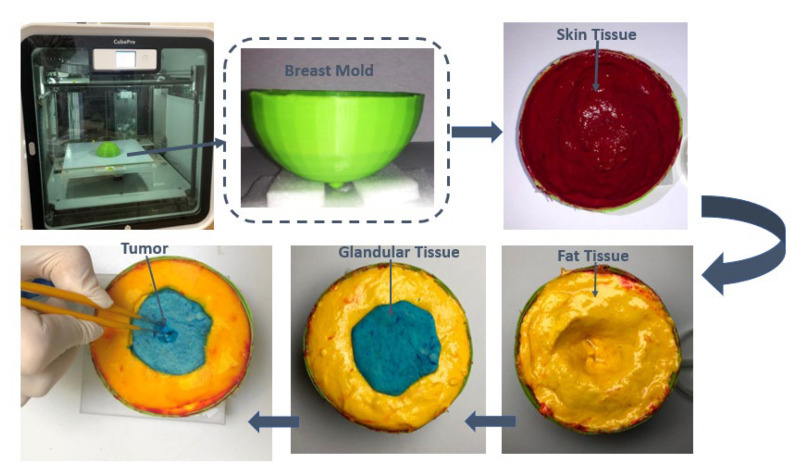
The final fabrication technique of the proposed breast phantom.

**Figure 19 materials-16-01496-f019:**
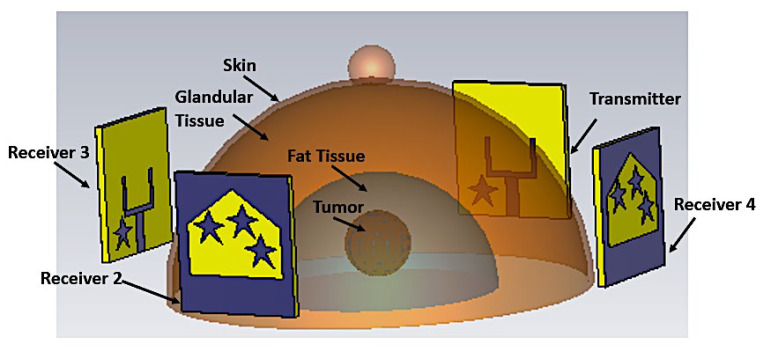
MWI simulation configuration.

**Figure 20 materials-16-01496-f020:**
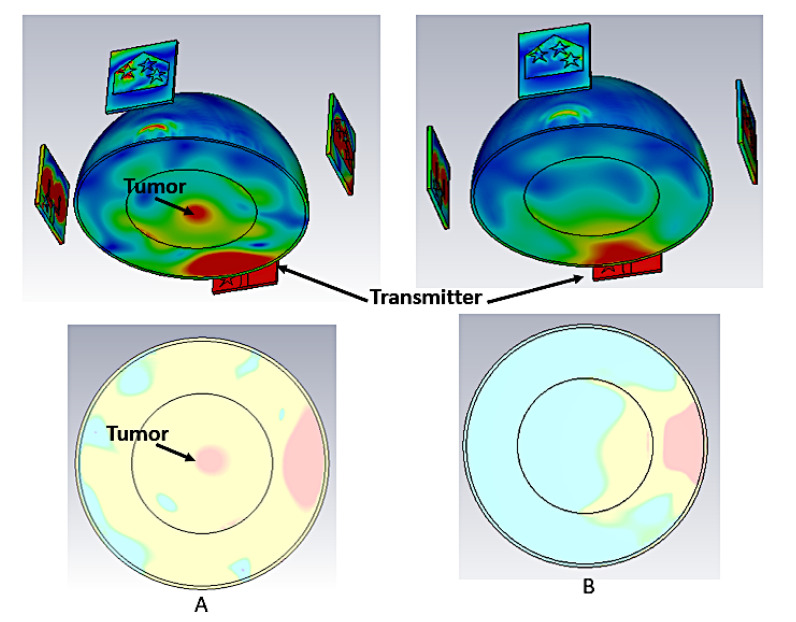
Results of the simulated setup (**A**) of an unhealthy breast and (**B**) of a healthy breast.

**Figure 21 materials-16-01496-f021:**
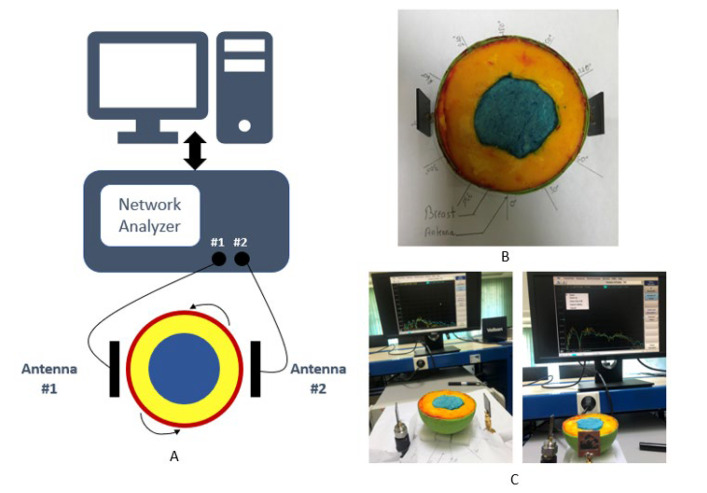
(**A**) The antennae surrounding the breast in this study’s data acquisition setup experiment, (**B**) photograph of the setup, and (**C**) sample of backscattering data measurements.

**Figure 22 materials-16-01496-f022:**
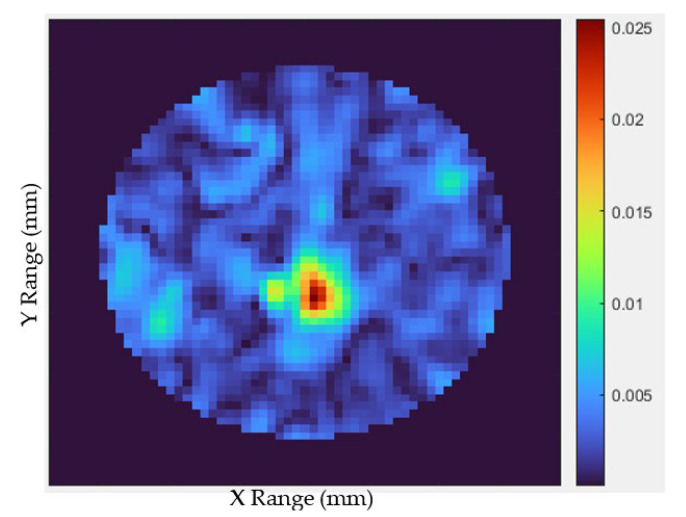
Results of the experimental realized MWI composed of the suggested antenna and a constructed artificial breast including a tumor.

**Table 1 materials-16-01496-t001:** Dielectric features of the breast tissue at 3 GHz.

Reference	Skin		Fat		Glandular		Tumor	
	εr	σ (S/m)	εr	σ (S/m)	εr	σ (S/m)	εr	σ (S/m)
[18]	38	0.94	5	0.17	45	1.6	56	1.7

**Table 2 materials-16-01496-t002:** Properties of various tissue layers of the breast phantom.

Material		Quantity				Purpose
		Skin	Fat	Gland	Tumor	
Sodium chloride (NaCl)		5 g	4 g	6 g	8 g	Improve the conductivity
Distilled water		20 mL	-	50 mL	100 mL	Increase the permittivity
Pure petroleum jelly		-	24 g	-	-	Modify the permittivity
wheat flour		10 mg	30 g	30 g	-	Thickener, Modify the permittivity
Olive oil		-	30 mL	-	-	Modify the permittivity
Powder dyes		(Orange)	(Yellow)	(Blue)	(Dark Blue)	distinguish the different components

**Table 3 materials-16-01496-t003:** Phantom comparison of UWB antennae in application of breast tumors.

Ref	Antenna				Fidelity Factor (%)	Phantom and Cancer Object	Setup Realization
	Type	Size (mm^2^)	Operating Frequency Range (GHz)	Gain (dBi)			
[4]	Quasi Log Periodic	50 × 40	2–5	4.6	Not reported	no	no
[23]	Vivaldi antenna	50 × 25	4–16	3.53	Not reported	Lab-made phantom	yes
[19]	Microstrip Antenna Composed of Parasitic Resonator	30 × 25	2.7–10.3	6.2	Not reported	no	no
[11]	UWB-Printed Rectangular-Based Monopole Antenna	40 × 36	2.7–11.4	6.6	88–93	no	no
[24]	Slotted Rectangular-patch	10 × 10	7.83–8.17	2.43	Not reported	no	no
proposed	Slotted antenna with four-star shape parasitic elements	29 × 26.6	3.8–10.1	6.8	91.6–91.2	Lab-made realistic heterogeneous phantom	yes

## Data Availability

All data are included within the manuscript.

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
