# Peer review of "Microwave Imaging Approach for Breast Cancer Detection Using a Tapered Slot Antenna Loaded with Parasitic Components"

_materials, 2023, doi:10.3390/ma16041496_

Round 1

Reviewer 1 Report

The authors presented a monopole with DGS for wideband operation and used it for breast cancer detection. The reviewer has the following comments.

1) Cite the reference for data shown in Figure 1.

2) Include the side view of the antenna in Figure 2. Also, improve the quality of the figure.

3) Mention the dielectric properties of the substrate used in the antenna fabrication.

4) The manuscript has serious issues with the technical soundness of the proposed design. Figure 3 shows the antenna response for the various design steps. The S11 (impedance matching) and the gain of the antenna is almost identical for all types of antennas (antenna without/with resonator and proposed design). This shows the parasitic patches (stars) are unnecessary. The primary radiator has a similar design concept (monopole with DGS) [https://doi.org/10.1016/j.aeue.2021.153612; https://www.mdpi.com/2079-9292/9/9/1366; https://doi.org/10.1007/s42452-020-04123-w]. Authors need to improve the design motivation as well as the contribution/novelty of this work. 

5) Include measured values of gain and efficiency to Figure 6. 

Author Response

Dear Reviewer

Please find the attachment, which is the response to your comments. Thanks

Best regards

Reviewer 2 Report

Dear authors, the paper is quite dense, containing both the antenna analysis and the performances when applied to the detection system, including the analysis of the pulse behavior for a broadband implementation.

Hover the presented design is not novel. Ref 18 discussed a very similar design. 

Moreover, the results presented in Figures 3 don't show any significant improvements due to the presence of the star resonators. There is a tiny improvement on the lower frequencies. Is that improvement so important for the targeted application?

Also, a discussion about size and positions of star resonators should be included, to justify the statement of "a novel antenna". Otherwise the paper should be focused differently, since the paper discuss more in detail the modeling of the breast.

There is another important detail that should be discussed. The need of a broadband response is clear, but also the radiation performances of the antenna system should be analyzed. Which kind of pattern and polarization are you targeting? Observing presented radiation patterns, the antenna behaves almost isotropically. Is that good?

Author Response

(The authors gave the same response as above.)

Round 2

Reviewer 1 Report

All the comments have been addressed well in the revised version. The updated manuscript can be accepted for publication.

Author Response

Dear Reviewer (1)

We are happy to know that you are satisfied with our revised manuscript. Thank you for your cooperation in making our paper look better.

Best regards,

Reviewer 2 Report

Dear authors, thanks for the proper answers to related comments about the paper. Most of the addressed points are now clarified. I think there is still a topic that would need some additional investigation. Since the frequency response of the antenna is related to time response (you know Fourier analysis), the reported improvements, in the time domain, seem uncorrelated to the fact that the antenna design with star resonators has very similar performance compared to the naked configuration.

I suggest also modifying the abstract: "In this paper, a novel wideband antenna is proposed for ultra-wideband microwave imaging applications". The antenna is not novel. Moreover, the design of star resonators (dimensions and placement) is made by CST itself, with no effort by the authors about some theoretical considerations supported by formulas and/or math.

Fig.4 lacks scale and units. Fig.7 lacks units.

row 294 "

tumor cells send more signal. As  
a result, the tumor cell has a high level of received signals"

That sentence need to be rephrased. A cell doesn't send any signal. A cell reflects/scatter an impinging wave. So it's the different dielectric constant of the malignant cells that makes the difference.

fig. 22 lacks units

Author Response

Dear Reviewer (2)

Please take a look at the attachment, which is the answer to your comments.

Best regards,
